# Overall Survival and Adjuvant Therapy in Women with Ovarian Carcinosarcoma: A Single-Institution Experience

**DOI:** 10.3390/diagnostics9040200

**Published:** 2019-11-22

**Authors:** Aaron Nizam, Bethany Bustamante, Weiwei Shan, Karin K. Shih, Jill S. Whyte, Antoinette Sakaris, Lisa dos Santos, Marina Frimer, Andrew W. Menzin, Alexander Truskinovsky, Gary L. Goldberg

**Affiliations:** 1Division of Gynecologic Oncology, Department of Obstetrics and Gynecology, Zucker School of Medicine at Hofstra/Northwell, Northwell Health, New Hyde Park, NY 11040, USA; bbustamante@northwell.edu (B.B.); kshih1@northwell.edu (K.K.S.); jwhyte@northwell.edu (J.S.W.); asakaris@northwell.edu (A.S.); ldossantos@northwell.edu (L.d.S.); mfrimer@northwell.edu (M.F.); amenzin@northwell.edu (A.W.M.); ggoldberg2@northwell.edu (G.L.G.); 2Department of Biostatistics, Northwell Health, New Hyde Park, NY 11040, USA; wshan@northwell.edu; 3Department of Pathology, Northwell Health, New Hyde Park, NY 11040, USA

**Keywords:** ovarian cancer, ovarian carcinosarcoma, adjuvant therapy

## Abstract

Background: Carcinosarcoma of the ovary (CSO) is a rare and aggressive variant of ovarian cancer. Due to the rare nature of the disease there is insufficient evidence to make recommendations regarding standard management and overall prognosis. Methods: An Institutional Review Board-approved study identified all our patients with CSO between January 2011 and May 2018. Demographic and outcome measures were abstracted from the medical records and tumor board files. Cox proportional hazard models, log rank tests, and comparisons of means were used to calculate significance (*p* < 0.05). Results: 27 women with CSO were identified. The median age at diagnosis was 65 years (range 48–91). Five women (18%) presented with early stage disease (Stage I or II) and 22 patients (82%) presented with late stage III or IV disease. Twenty patients (74%) received intravenous platinum-based combination chemotherapy. Seven patients did not receive chemotherapy during their treatment course. The median overall survival was 23 months (range 2–68 months). Overall survival was not significantly worsened by the stage of disease at diagnosis. There was no difference in survival based on the age at diagnosis, tobacco status or ethnicity (*p* > 0.05). Conclusion: This is one of the largest single institution experiences with CSO. The majority of our patients presented with advanced stage disease and received adjuvant platinum-based chemotherapy after cytoreductive surgery. The median overall survival of 23 months was not affected by the stage of the disease. The optimal management of this rare disease needs further study with collaborative, prospective multi-institutional trials.

## 1. Introduction

Ovarian carcinosarcoma was once thought to be an extremely rare form of ovarian cancer representing between 2–6% of all primary ovarian malignancies [1]. These tumors are characterized by a mixture of malignant epithelial and stromal cells. It is the combination of both components that differentiates CSO from the more common epithelial ovarian carcinomas. Most patients are diagnosed with advanced stage disease. Although maximal cytoreductive surgery followed by chemotherapy is often utilized in the management of this disease, there are no randomized clinical trials supporting this approach to the treatment of CSO. Many patients are treated with a combination of platinum and taxane-based chemotherapy [2]. Most patients have disease recurrence within one year and survive approximately two years after the initial diagnosis [3,4]. A review of all women from 1998–2009 with CSO from the Surveillance, Epidemiology, and End Results (SEER) Program data demonstrated a poorer prognosis than those with high-grade serous ovarian cancer [5].

There are three current hypotheses regarding the development of carcinosarcoma. The conversion theory states that the sarcomatous element of the tumor is formed from the evolution of the carcinomatous element of the tumor. The combination theory suggests that both components are derived from a single precursor cell that undergoes divergent differentiation. The third and least likely is the collision theory, which suggests that the carcinomatous and sarcomatous elements develop independently of each other. Current evidence suggests that both elements are derived from a single metaplastic precursor. There is no obvious inherent genetic mutation in women with CSO. CSO is diagnosed pathologically with evidence of invasive carcinoma and sarcoma derived from the ovary. They are staged identically to epithelial ovarian cancers [3,4,5].

Given the rarity of the disease, it has been difficult to extrapolate firm recommendations for the management and treatment of CSO. The majority of published reports consist of single-institution studies, most of which show a worse progression free survival and overall survival. As there are still fundamental questions regarding the optimal management of this rare form of ovarian cancer, we conducted a retrospective systematic review of all women treated at our institution from 2011–2018.

## 2. Materials and Methods

An Institutional Review Board-approved study identified all patients with CSO treated at Northwell Health institution from January 2011 to May 2018. Data reviewed included demographic information, surgical, clinical and adjuvant therapy information including dates of follow up. All patients diagnosed prior to 2014 were restaged using the International Federation of Gynecology and Obstetrics (FIGO) 2014 staging system. Patients were included if they were treated by a gynecologic oncologist or medical oncologist at our institution during the study period with a confirmed pathologic review of CSO by a fellowship-trained gynecologic pathologist at our institution (Figure 1). Primary surgery or first-line chemotherapy were required to take place at our institution for inclusion.

Our primary endpoints were overall survival (OS) and progression-free survival (PFS). OS was calculated from time of diagnosis to time of death or last follow-up if the exact date was unknown. Progression-free survival was calculated from the time of diagnosis to the date of diagnosed progression or recurrence. Recurrence was defined a tissue diagnosis confirming CSO after a period of time where the malignant cells were not detected. Platinum sensitivity was defined as a recurrence of disease six months or more after completion of platinum-based chemotherapy. Results were analyzed and statistical analysis was performed using R Statistical Software (Foundation for Statistical Computing, Vienna, Austria). A chi-squared test was performed for categorical variables and *t*-test was performed for continuous variables. Kaplan-Meier curves were plotted for PFS and OS and log-rank test was used to compare PFS and OS between groups. A *p*-value of < 0.05 was considered statistically significant.

## 3. Results

There were 34 patients diagnosed with CSO at our institution between January 2011 and May 2018 (Table 1). All patients were identified through our institutional prospectively collected database. Four patients presented with recurrent disease after prior treatment at an outside institution, 2 patients had no documented follow-up and 1 patient was diagnosed with a concurrent malignancy. Twenty-seven patients were eligible for inclusion in the final analysis.

The median age at time of diagnosis was 65 (range 48–91 years). Eighteen of the 27 patients diagnosed with CSO were white, with 5 African American patients and 4 Asian patients. The majority of the women in our cohort were diagnosed with advanced-stage disease (83%). Two patients (7%) were diagnosed with stage I disease, three patients (11%) with stage II disease, 13 patients (48%) with stage III disease and nine patients (33%) with stage IV disease. Eleven patients (41%) presented with bilateral ovarian masses, nine (33%) with a left sided adnexal/pelvic mass and seven (26%) with a right-sided adnexal/pelvic mass.

Of the 27 patients analyzed, 20 received chemotherapy. Two patients underwent neoadjuvant chemotherapy with 3 cycles of carboplatin/taxol followed by optimal cytoreductive surgery and 3 cycles of carboplatin/taxol in one patient and suboptimal cytoreductive surgery with multiple failed chemotherapy regimens in the other. Seventeen patients underwent adjuvant chemotherapy all but one of whom received a platinum-based regimen. Fourteen of the 17 patients received carboplatin/taxol, one patient received carboplatin/taxol/avastin, one patient received carboplatin/ifosfamide and one patient received single-agent carboplatin due to medical comorbidities. One patient was treated with adjuvant intravenous and intraperitoneal (IV/IP) therapy. Eleven of 20 patients who received chemotherapy had platinum sensitive disease. Two patients were unable to complete six cycles of carboplatin/taxol due to adverse reactions and were transitioned to second-line agents.

Two patients underwent adjuvant radiation therapy. One patient was not a surgical candidate and had evidence of peritoneal and bilateral axillary disease. She received 4000 centigray (cGy) to the pelvis and 3000 cGy to both axillae. The patient is currently alive with disease. The second patient had optimally debulked stage IIIC disease with six cycles of carboplatin/taxol with an isolated recurrence in the right axilla 11 months later and received 3000 cGy to the axilla and is currently alive with no evidence of disease.

The median survival for our cohort was 23 months (range 1–65 months, Figure 2). The three-year overall survival was 22%. No surviving patients were lost to follow-up at the three years. There was no statistically significant difference in overall survival based on stage of disease at diagnosis (Figure 3). Stage of disease was not predictive of recurrence (*p* = 0.93). Patients receiving adjuvant chemotherapy had a significantly longer overall survival than those that did not receive adjuvant chemotherapy. There was also no difference in survival based on those with bilateral ovarian masses versus unilateral masses (*p* = 0.69). Overall survival was significantly worse in those with a diagnosed recurrence (12.8 vs. 29.3 months, *p* < 0.05). The median PFS for our cohort was 9 months. All 11 patients that recurred were treated with carboplatin/taxol. Twelve of 14 (70.6%) of the patients that self-reported never using tobacco were diagnosed with disease recurrence compared to five of 11 (45%) current or former smokers (*p* = 0.028). Seventeen of 27 patients (63%) were diagnosed with recurrence during the study period. Of all patients receiving adjuvant treatment, eleven were platinum sensitive, three were platinum refractory and six were platinum resistant. Demographics based on recurrence status are presented in Table 2.

## 4. Discussion

CSO represents a rare and aggressive subtype of ovarian cancer. Patients tend to present at an older age with large ovarian masses with hemorrhagic components and necrosis when compared with serous carcinoma of the ovary [4]. The majority of patients present at an advanced stage and are treated with chemotherapy with platinum and taxane [6]. Despite CSO accounting for roughly 2–6% of all diagnosed ovarian cancers, we still do not know the optimal treatment for these women with CSO. They tend to recur earlier and have a decreased OS when compared to other subtypes of epithelial ovarian cancer.

Although other studies have demonstrated a survival benefit for those diagnosed with early-stage disease, our cohort did not demonstrate improved survival or a decrease in recurrence. A National Cancer Database analysis also found a decrease in both cancer-specific and OS when compared to serous ovarian carcinoma across all stages [7]. Similar results were found in a review of 1334 women with CSO from patients in the SEER database from 1998–2009 in which nearly all factors were independently predictive for worse cancer-specific survival for those with CSO compared with serous carcinoma of the ovary [5]. Despite this data, we still manage both disease entities with the same chemotherapy.

It has been difficult to establish a standardized treatment protocol for CSO due to the rarity and the aggressive nature of the disease. A prospective Gynecology Oncology Group (GOG) trial treating 136 patients with CSO with cisplatin 50 mg/m^2^ every three weeks until disease progression or unacceptable toxicity showed about a 20% response rate [8]. A retrospective review of 26 patients demonstrated a statistically significant increase in survival of 26 months when comparing ifosfamide or carboplatin/taxol to other regimens [9]. Ifosfamide based chemotherapy for adjuvant treatment has been shown to also have about a 20% response rate [10]. A similarly sized cohort treated at Memorial Sloan-Kettering Cancer Center found that treatment with carboplatin/taxol may be a suitable first-line chemotherapy regimen with a median survival of 43 months in 30 treated patients [11]. Due to the rarity of CSO, the GOG stated that we will likely need to extrapolate treatment options from those with uterine carcinosarcoma as it will be increasingly difficult to set up and complete a randomized controlled trial [7]. A case-control study comparing treatment in 50 women with CSO to 100 controls found that with platinum and taxane based chemotherapy those with CSO had decreased PFS by 5 months and decreased OS of 17 months [12]. Given the overwhelming evidence that ovarian carcinosarcoma has a worse response to the standard treatment for epithelial ovarian cancer, alternate regimens need to be explored to improve post-operative PFS and OS. A recent case report showed a significant decrease in tumor burden in a 67-year-old patient with stage IIC CSO when treated with the PARP inhibitor olaparib [13]. Another study reported that patients with PD-L1 expressing tumor cells had a poorer survival than those with negative PD-L1 tumors, suggesting that there may be a role for immunotherapy targeting this pathway [14].

Within our cohort we did not find a decrease in recurrence rates based on age, ethnicity, family history of cancer, tumor laterality or stage of disease. We did find significantly more recurrences in those who were self-reported as “never smokers” compared to “current” or “former smokers”. A biopsy proven recurrence did decrease OS by 10 months within our cohort. We also found no difference in the frequency of recurrences in those treated with chemotherapy versus those without chemotherapy. OS was improved with adjuvant chemotherapy, however, only five patients did not pursue chemotherapy and two of those five declined any form of treatment due to prior medical conditions and poor performance status. It is interesting that two of our cohort had metastases to the axilla which is unusual for epithelia ovarian cancer.

We recognize the limitations of this retrospective cohort analysis. However, we believe that this data adds to the current knowledge of this rare disease. It is clear that CSO requires a different management approach compared to ovarian epithelial cancers. Extrapolation form uterine carcinosarcoma does not appear to affect the management of CSO. The size of our cohort is an obvious limitation. It makes the statistical analysis difficult to interpret and extrapolate.

The nature of CSO makes it a difficult disease to study and treat objectively. Due to the rarity of the disease, the majority of data presented originate in case reports and small retrospective reviews [3,9,11,12,15,16]. It has become clear that CSO is more aggressive than serous ovarian carcinoma with worse overall outcomes at any stage. There is a definite need to expand our efforts to initiate larger randomized controlled trials and genomic tumor analysis for CSO in order to improve our patient’s overall survival and quality of life. This will require a multi-institutional collaboration and increased enrollment on clinical trials for women with this rare and aggressive disease.

## Figures and Tables

**Figure 1 diagnostics-09-00200-f001:**
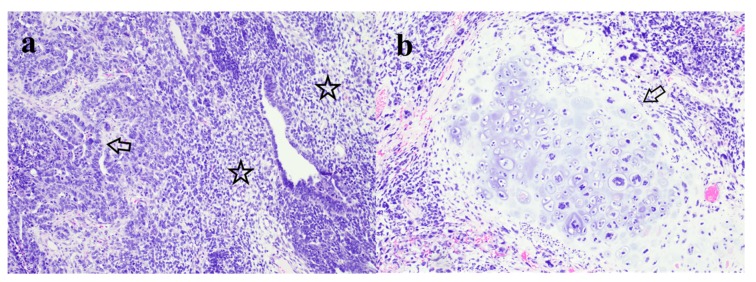
Malignant mesodermal mixed tumor (carcinosarcoma) of the ovary. (**a**) Juxtaposed malignant epithelial (arrow) and mesenchymal (stars) elements of the tumor. Hematoxylin and eosin, original magnification ×100. (**b**) Focal malignant heterologous (chondroid) stromal differentiation (arrow). Hematoxylin and eosin, original magnification ×100.

**Figure 2 diagnostics-09-00200-f002:**
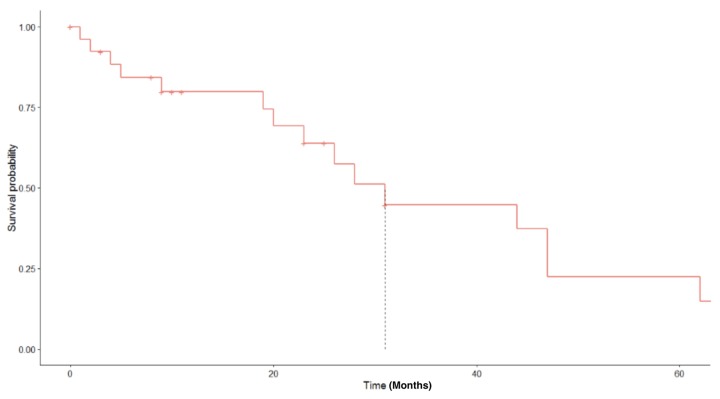
Overall survival for entire patient cohort.

**Figure 3 diagnostics-09-00200-f003:**
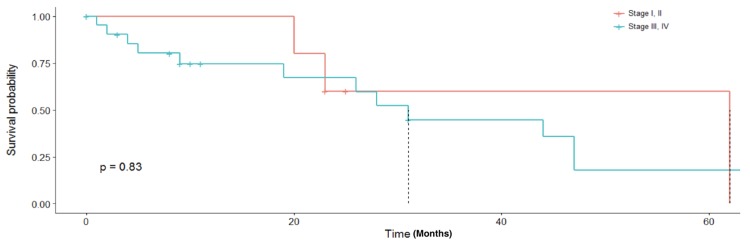
Overall survival by stage. Kaplan–Meier curve comparing early stage (I and II) to advanced stage (III and IV).

**Table 1 diagnostics-09-00200-t001:** Demographic Information.

Characteristics	Patients (%)	*n*
Mean Age (SD)	64.7 (17.1)	27
Race		27
Asian	4 (14.8)	
Black	5 (18.5)	
White	18 (66.7)	
Smoking Status		25
Current smoker	1 (4.00)	
Former smoker	10 (40.0)	
Never smoker	14 (56.0)	
Family History: Cancer		20
No	7 (35.0)	
Yes	13 (65.0)	
Laterality		27
Bilateral	11 (40.7)	
Unilateral	16 (59.3)	
Stage		27
Early Stage: I, II	5 (18.5)	
Advanced Stage: III, IV	22 (81.5)	
Adjuvant Chemotherapy		27
IV	19 (70.4)	
IV/IP	1 (3.7)	
None	7 (25.9)	
Recurrence Status		27
No recurrence	10 (37.0)	
Recurrence	17 (63.0)	
Platinum Sensitive	11 (40.7)	
Optimal Cytoreduction	20 (75.0)	
Suboptimal Cytoreduction	7 (26.0)	

**Table 2 diagnostics-09-00200-t002:** Demographic information based on recurrence status.

	No Recurrence *n* = 10	Recurrence *n* = 17	*p*-Value
Age	69.8 (11.9)	61.6 (19.2)	0.185
Race			0.215
Asian	0 (0.00%)	4 (23.5%)	
Black	3 (30.0%)	2 (11.8%)	
White	7 (70.0%)	11 (64.7%)	
Smoking Status			0.028
Current smoker	0 (0.00%)	1 (5.88%)	
Former smoker	6 (75.0%)	4 (23.5%)	
Never smoker	2 (25.0%)	12 (70.6%)	
Family History of Cancer			0.613
No	3 (50.0%)	4 (28.6%)	
Yes	3 (50.0%)	10 (71.4%)	
Mass Laterality			1.000
Bilateral	4 (40.0%)	7 (41.2%)	
Left	3 (30.0%)	6 (35.3%)	
Right	3 (30.0%)	4 (23.5%)	
Stage			1.000
Early Stage: I, II	2 (20.0%)	3 (17.6%)	
Advanced Stage: III, IV	8 (80.0%)	14 (82.4%)	
Adjuvant Chemotherapy			1.000
IV	8 (80.0%)	11 (73.3%)	
IV/IP	0 (0.00%)	1 (6.67%)	
None	2 (20.0%)	3 (20.0%)	
Overall Survival (Months)	29.3	12.8	0.015

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
