# Peer review of "Overall Survival and Adjuvant Therapy in Women with Ovarian Carcinosarcoma: A Single-Institution Experience"

_diagnostics, 2019, doi:10.3390/diagnostics9040200_

Round 1
Reviewer 1 Report
The authors have shown overall survival of CSO patients with respect to different stages of the disease. They also presented the demographic information of patients, however few important points are missing in the manuscript.
Line 18 – “27 women with CSO were identified.”How CSO patients are identified?
Line 56 – What are the “confirmed pathologic review of CSO”?
As primary surgery is the major inclusion criteria for patients, the authors must have access to tissue biopsy samples. H&E images of CSO patients are expected in the manuscript. How the histology of a CSO specimen is different from ovarian cancer? The authors should explain in detail what is “carcinosarcoma of the ovary (CSO)”, how it is originated, any specific gene mutation and how it is different from ovarian cancer. The introduction must be elaborated.
Line 28 & 29 – “The median overall survival (23 months) was not affected with adjuvant therapy”. Line 99 &100 – “Patients receiving adjuvant therapy had a significantly longer survival than without adjuvant therapy”.
The results should be consistently presented throughout the manuscript.
Figure 3 is missing in the manuscript.
The adjuvant therapy data should be shown in the manuscript.
In figure 2 authors have shown overall survival of CSO patients by different stage (I/II/IV). How authors categorized CSO patients with different stages. H&E images of different stages of CSO must be shown in the manuscript.
Author Response
Dear Reviewer,
Thank you for your comments and suggestions. Please find the below response:
Line 18-27: Identification of patients was explained further and pathologic review was described.
A histologic sample was provided as Figure 1.
A paragraph was added to the introduction to explain the theories of the origin of ovarian carcinosarcoma.
There is no specific gene mutation that has been discovered thus far. A sentence was added into the first line of the introduction to describe the difference between carcinosarcoma of the ovary and epithelial ovarian carcinoma.
Line 28-29, 99-100. The distinction was corrected.
Figure 3 is present with description and citation currently.
Stage of disease for CSO is the same as for epithelial ovarian cancers. The staging is performed surgically. There is no difference on H&E from different stages as stage is related to the spread to sites other than the ovary. This distinction is now described in the paper.
We hope that you find these changes satisfactory.
Sincerely,
Aaron Nizam MD
Reviewer 2 Report
This study aims to present a brief clinical description of the rare ovarian carcinoma presentation from a single-institution. Because different institutions may use different types of treatment or patient management, a single institution study may be less bias to compare clinical factors in a cohort. This article could contribute valuable data towards the description of this disease. However, there are important issues that the authors must address:
Major Comments:
The studied cohort is very small and statistical significance may not be reached due to the size of the cohort rather than a lack of association between factors. The authors should verify and provide the statistical power for each analysis. As the authors often compare CSO and high-Grade Serous carcinoma of ovary (HGSOC), they should also show a comparative analysis of their CSO cases with some HGSOC from their institution. Table 1 could be a nice place to show such comparison. Comparative survival of CSO and HGSOC could also be shown in Figure 1.
Minor Comments:
Lane 42-43: authors should provide a reference for this statement. Materiel and Methods: authors should provide a definition of recurrence and platinum sensitivity The number or % of censored patients (patients without follow-up) at 3 year should be indicated. The statistical test used to compare continuous variable is not indicated (i. age). Table 1: residual disease should also be indicated among the demographic factors Table 1: platinum-sensitive patients should also be indicated among the demographic factors and in association with recurrence in Table 2 Lane 125: please provide references to these statements In the discussion section, authors should clearly mention the size of the cohort as a strong limitation in their study.
Author Response
Dear Reviewer,
Thank you for your comments and suggestions. Please find the below response:
Due to the small number of patients in our cohort it was difficult to reach statistical significance. This is mentioned in the conclusion as a limitation to our study.
As CSO is a rare variant of epithelial ovarian cancer (for which HGSOC is the most commonly found histology) we found it necessary to compare the subtypes. Given the significant data on HGSOC we believe that historical comparisons were more important as direct comparison within our cohort of patients would divert focus away from this extremely rare subtype.
Line 42-43: A reference was added. We now define both recurrence and platinum sensitivity.
A line was added (166) mentioning the follow up at 3 years (22% of the cohort, but 100% of patients alive at that time).
The statistical test used for continuous variables was added. Platinum sensitivity and residual disease was added to table 1.
Sample size mentioned as strong limitation to our study.
We hope that you find these changes satisfactory.
Sincerely,
Aaron Nizam MD
Round 2
Reviewer 1 Report
The authors have improved the manuscript. But it can be better,
H&E images could have been better. I didn't find the adjuvant therapy results.Reviewer 2 Report
1) The statistical power for each analysis is not indicated . Authors have not responded to the major comments of the reviewer.
2) A comparative analysis between HGSOC and CSO is not shown. Authors do not respond to the reviewers' major comments. Since the authors based their comparison with historic data (i.e scientific literature) what is the point of this study ? Description and comparative outcome of CSO and HGSOC have already been published in large multi-institutional studies.
4) The rate of survival at 3 years (22%) is very different from the rate of recurrence (60%), which mean there is a high risk of dying from the another cause than CSO. It is then not clear what is the cause of death in CSO patients of this institution.
3) Finally, how can the survival at 3 years be 100% ? Does that mean all patients diagnosed since 2016 have died, while the survival rate is 22% ? In the figure 1 and figure 2, it seems there are at least 5 patients without follow-up at 3 years.